# Econometric Model for Readjusting Significance Threshold Levels through Quick Audit Tests Used on Sustainable Companies

**Veronica Grosu [1,*], Dorel Mateș [2], Monica-Laura Zlati [1], Svetlana Mihaila [3], Marian Socoliuc [1], Marius-Sorin Ciubotariu [1] and Simona-Maria Tanasă [1]**

[1] Department of Accounting, Auditing and Finance, "Ștefan cel Mare" University of Suceava, 13 Universitatii Street, 720229 Suceava, Romania; sorici.monica@usm.ro (M.-L.Z.); marian.socoliuc@usm.ro (M.S.); marius.ciubotariu@usm.ro (M.-S.C.); tanasa_simona19@yahoo.com (S.-M.T.)

[2] Department of Accounting, West University of Timișoara, 4 Vasile Parvan Boulevard, 300223 Timișoara, Romania; dorel.mates@e-uvt.ro

[3] Department of Accounting, Academy of Economic Studies of Moldova, 2005 Chișinău, Moldova; svetlana.mihaila@ase.md

* Correspondence: veronica.grosu@usm.ro; Tel.: +40-743-421-464

**Abstract:** Given the present-day economic situation, which is characterized by economic destabilization as a result of the pandemic crisis, auditors are facing the issue of establishing materiality, which is partly based on the fact that a certain level of financial misstatement influences the decisions of the involved parties. The aim of the present study is to suggest an econometric model for readjusting significance threshold levels through quick audit tests used on sustainable companies. The main objectives of the study are to emphasize the causal relationship between the manifestation of constant errors in financial reports and the inconsistency of audit opinions, as well as to put into practice the causal relationship that exists between the improvement of the audit function and sustainability itself, given companies' crisis situation. In this particular context—based on the entire sample of companies listed in the Bucharest Stock Exchange (BVB), Bucharest Exchange Trading Plus category (BET Plus)—we estimated a number of financial indicators between 2009 and 2018 so that we could determine the materiality of accounting errors identified by auditors in order to express an opinion regarding the reliability and accuracy of financial reporting. The study's key findings show that, given the economic crisis, the significance threshold level is a volatile test and it needs to be reconsidered taking into account the decline in the quality of reporting and, indirectly, the disclosure of financial information. From a holistic point of view, we believe that our study will have a significant impact on both practitioners and regulatory entities by shifting the qualitative approaches of analysis itself towards key prudential regulations stipulated by International Standard on Auditing (ISA) 320, ISA 450 and ISA 700. The study also highlights the process of refining information sources that can impact the significance, understanding and materiality of business decisions.

**Keywords:** materiality threshold; audit procedures; financial efficiency; economic sustainability; misstatement

## 1. Introduction

As a result of the coronavirus pandemic, companies' current economic situation is influenced by economic blockage. This situation has resulted in a series of profound disturbances to business activities within the European Union. These disturbances have led to a series of micro- and macroeconomic phenomena that have increased companies' vulnerability by destabilizing exchange rates, by influencing

their market shares as a result of a reduction in consumption itself, and by having a negative effect on the demand for commodities and services. From this point of view, the roles of both financial auditors and financial experts as guarantors of the smooth running of businesses and of accurate financial reporting are significantly hindered and face challenges that characterize an economic crisis similar to that of 2008–2009. In particular, during these times, due to the existing uncertainty, users/stakeholders request more information that is mainly related to auditors [1,2].

From an economic perspective, the level of quotations on the Bucharest Stock Exchange (BVB) went through a regression phase at the beginning of the coronavirus pandemic this year in March. Most companies have experienced a drastic decrease in the value of quotations. Generally speaking, the stock exchange experienced a decrease in its calculated composite indices. It is worth mentioning that at the time of our study, both financial statements and audit reports for 2019 have not yet been made public by the companies that are part of our studied group. Audit missions were conducted based on the international audit standards (ISA), which have been adopted in Romania, and they were put into practice in several stages, including analysis, evaluation, reinterpretation and formulation of a certified audit statement. In the current situation, we believe that these statements should comprise an extra reinterpretation section as part of the financial reporting, as well as an overall reconsideration of significant information and significance thresholds. Moreover, we need to take into account economic aspects such as the reduction in the need for consumption, rise in inflation, market destabilization, rise in the number of companies facing bankruptcy, etc. [3]. At the same time, from the point of view of their economic or financial relationships, companies face uncertain times due to the accumulation of overdue tax liabilities that will affect company activities in the short and medium term [4,5]. Another aspect contributing to the overall picture is the growth of unemployment and of the social pressure that employers need to experience so that they are able to keep their qualified staff and ensure the smooth running of their future activities [6].

These conditions may lead to a global economic crisis, which thus affects auditors' activities. Auditors have a tendency to diminish their charges during critical times, even though they have an increased workload due to an increasingly competitive audit market [7]. Moreover, the outcome of a crisis is reflected by the rise in the number of qualified opinions expressed, because of the number of clients that find themselves in a difficult situation [8,9].

Thus, there is a need for a well-documented study regarding the re-evaluation of financial statements by using audit missions, which are completed according to the present-day uncertain conditions affecting companies' activities.

The aim of this study is to suggest an econometric model for readjusting significance threshold levels through quick audit tests used on sustainable companies. The main objectives of the study were to emphasize the causal relationship between the manifestation of constant errors in financial reports and the inconsistency of audit opinions, as well as to put into practice the causal relationship that exists between the improvement of the audit function and sustainability itself, given companies' crisis situation.

Taking into account the present-day financial and economic situation and its external importance, we believe it is necessary for involved parties to oversee the information that needs to be disclosed in financial reports [10] (in terms of ongoing activities, financial risks, inspections of asset depreciation and uncertainty associated with the use of forecasts). This information must be real and must not have been tampered with by the reporting company. Improving the quality of audit becomes even more important during periods of financial crisis (i.e., liquidity issues), when companies are encouraged to improve their financial reports in order to attract funding. In this context, auditors tend to be more conservative in terms of the quality of the information that is disclosed in financial reports.

Given the impact of the present-day worldwide situation, which is influenced by an economic crisis that has highlighted certain weak points of audit practices the purpose of the research team was focus on suggesting meaningful analysis tools based on statistical and econometric methods that are able to improve audit results by using quick audit tests on the significance threshold. We have

taken into account the European pro-sustainability approach. Thus, our study has focused on listed companies that make use of the sustainability principles that have been embraced throughout Europe.

From a methodological point of view, the research was based on financial information from companies that were listed in the Bucharest Exchange Trading Plus category (BET Plus) on the Bucharest Stock Exchange. This information was integrated in a matrix database, on which were applied a few series of statistical frequency tests and data homogeneity tests. The final result was a molded database to which we applied statistical procedures in order to obtain the suggested econometric model.

The following research objectives supported the achievement of the aims of the present study: (1) demonstration of the causality relationships between the continuous errors that are part of financial reporting and the inconsistency of audit opinions; (2) demonstration of economic sustainability for companies that constantly improve their audit-based activities; (3) demonstration of economic resilience during a crisis for companies whose economic sustainability is improved by using audit techniques.

The aim of this study is to design a quick test for re-evaluating significance threshold based on the analysis of the financial performance of companies listed on the BVB in the BET Plus liquidity section.

Our findings contribute to the following aspects: firstly, we examined the connection between significance threshold, quality of accounting information and foundation of audit opinions given the economic crisis [11–13]; secondly, the study suggests an analysis of the specialty literature, which continuously emphasizes the existence of certain malfunctions of audit policies that are regulated by International Standard on Auditing (ISA) 320, ISA 450 and ISA 700 [14]. Several researchers [15,16] have asserted that "the use of the materiality itself brings about a certain audit risk level from the point of view of the fact that in the field of accounting the representation of a true and qualitative image is tightly connected with its type, intensity and dimension". Furthermore, the specialty literature [17,18] has come to the conclusion that there is an inversely proportional relationship between significance threshold and audit risk, which means that the higher the materiality level, the lower the audit risk and vice-versa.

Finally, based on the econometric model, we discovered that, during an economic crisis, the level of the materiality threshold for audit procedures needs to be at least 30% in order to ensure the validity of the audit solution.

An extra certification during the planning stage is the establishment of two thresholds—a performance materiality and a clear trivial threshold (i.e., the tolerable error). It is worth mentioning that even in this circumstance, the auditor will use his/her professional judgment. In practice, the establishment of the performance materiality is done based on using the value of the significance threshold for a 60–70% share. The trivial threshold is estimated based on using a 1% to 10% share for the value of the significance threshold [19].

This study comprises five sections. Section 2 presents a theoretical framework that helped us formulate the tested hypotheses. It also includes a review of the literature on materiality associated with the audit risk, quality audit or auditor's professional judgment. Section 3 sets out the research methodology of our study, including the data sources and the details of the developed QATRMT model (i.e. the quick audit test for readjusting materiality threshold). Section 4 presents the results of our research. Section 5 provides a conclusion of the findings of our study, as well as the limitations of our research and possible future trends.

Our study is useful for professionals from the point of view of making use of best practices when formulating a proper certified audit opinion, given the present situation of the activities of the company itself. Our study is also useful for all interested parties, especially clients, investors and sponsors, by allowing them to better understand the importance of the qualitative nature of materiality itself from the point of view of the auditors' opinions. Moreover, the findings of our research are useful for the listed companies as they can contribute to a rise in stakeholders' trust in both financial and non-financial reporting, which will determine their decisions regarding those listed companies.

## 2. Theoretical Background

### 2.1. A Brief Overview of the Scientific Field

In the context of today's economy, as far as audit is concerned, the concept of significance threshold (materiality audit) in audit has been widely discussed in the international literature [20]. This is the result of big international financial scandals, of the global financial crisis of 2008, and of the tendency to want to clarify this concept.

The concept of significance or materiality threshold has been previously used in the field of accounting in order to elaborate financial statements, being defined as the significance of the elements that are part of the statement's structure directly linked with the interests of the main stakeholders [21]. Secondly, the notion of significance is also used in audit activity, but with a less clear meaning for stakeholders due to the fact that the significance threshold is in a close relationship with the value of the final balances that are reported in the financial statement [22,23], on the one hand, and in the corresponding costs for the audit services, on the other hand (the actual value of the significance threshold depends on the auditor's professional reasoning). Given the opinion of Houghton et al. [21], we believe significance itself is a far more complex issue. Nowadays, it is used as part of sustainability reporting, namely, integrated reporting. As far as sustainability reporting is concerned, the significance threshold is the principle setting up the very nature of the significant subjects that are viewed as being the key elements to be reported by a company. The significance itself helps in evaluating a company's economic, social and environmental impact within a certain market sector in which it operates [24]. Moreover, a piece of information that is reported by a certain company is considered to be useful when it has a certain impact on the stakeholders' decisions [25]. This is the reason why it needs to be present as part of the sustainability report [26]. It must be highlighted that, given the context, the significance needs to be regarded from the point of view of the stakeholders' demands, namely, it refers to the information that they look forward to finding in this kind of report.

Based on the International Integrated Reporting Council (IIRC) [27], this significance refers to the information that has a major impact on a company's ability to generate value in a short, medium or average period of time as well as to the very elements that need to be included in an integrated report.

According to the requirements of the Global Reporting Initiative (GRI) [26] and the IIRC [27], respectively, there are studies that either define the concept of significance or highlight the differences that exist between the significance used when reporting on sustainability and that used when designing financial reporting [28–30].

The significance threshold is defined as part of the general framework for designing and presenting financial statements of the International Accounting Standards Board (IASB) [31], where it is clearly stated that "information is material if its omission or misstatement could influence the economic decisions of users taken on the basis of the financial statements". The significance threshold depends on the size of the element or of the error itself, which is viewed in relationship with the particular conditions of the omission or the inaccurate report. Thus, the significance threshold imposes more of a limit instead of being a primary qualitative trait characterizing the information in order for it to be useful.

According to the Financial Accounting Standards Board (FASB), the significance reveals the importance of an omission or of an erroneous presentation of the financial and accounting information, which, from the point of view of the overall circumstances, can have an impact or can shift the way of thinking of a reasonable individual based on that information [32]. The significance is actually regarded from the same point of view as that of the IASB.

The Australian Auditing Standard (ASA) 320 defines the significance as a key element when planning an audit mission, and its evaluation is based on the following elements: the establishment of a preliminary significance threshold and its further evaluation, the identification of the significance's qualitative and quantitative factors, and the evaluation of the audit errors and risks [33].

According to ISA 320, the significance threshold is defined as "the level, the size of a sum over which the auditor thinks that a mistake, an error or an omission may affect both regularity and accuracy of the yearly accounts, and the true image of the final result, of financial statement and of the company's assets" [18].

As far as the topic of study is concerned, we used "quality audit" and "significance audit" as search parameters on the Web of Science (WoS) platform for 2000–2020 t. We thus found 284 articles and proceedings papers for "quality audit" as shown in Figure 1 based on the publishing year, research field and continent. We noticed that the number of published studies on this topic varies during the analyzed period, reaching the maximum number of papers published in 2017. The field of research that the majority of the studies belong to is business finance; however, this concept can be found in over 20 fields of research. The highest number of studies published on this topic is in Europe where England occupies the top position with a number of 28 studies, followed by Asia (85) and America (71) where the USA leads with 61 studies published between 2000 and 2020.

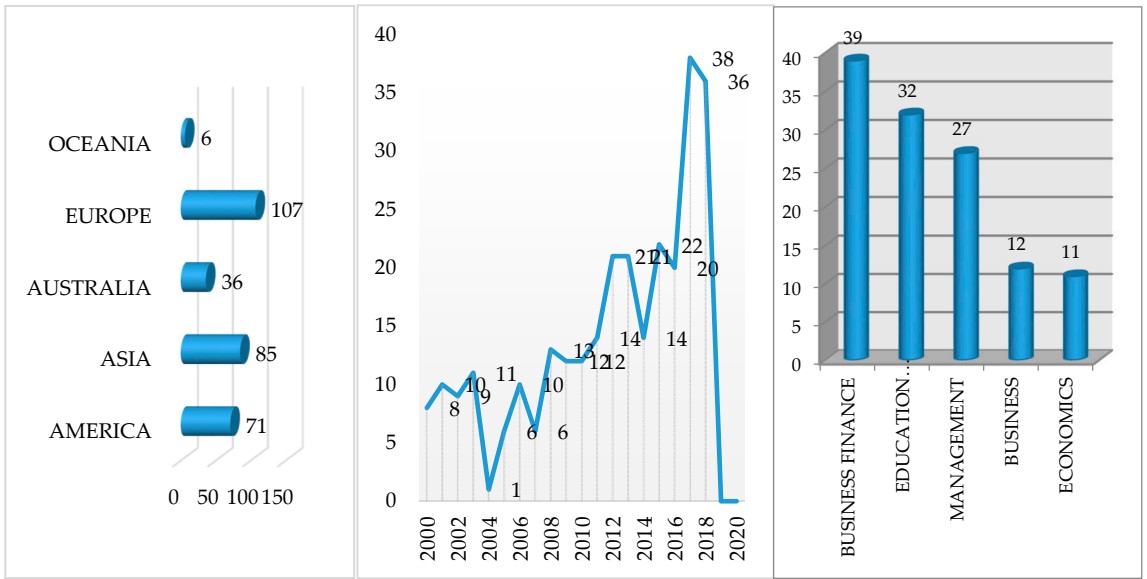

**Figure 1.** The profile of research on quality audit between 2000 and 2020. Source: developed by the authors on the Web of Science (WoS) database.

In the study undertaken by Al-Qatamin and Salleh [34], the authors reiterates the existence of a variety of research in terms of the "quality audit" concept in numerous fields of research, due to the fact that there is no universal definition of the term and different authors provide their own definition for it. Moreover, the authors made a summary of indicators of the quality of an audit, such as professional skepticism, professional expertise, professional experience and risk assessment, which have a direct impact on this concept.

In terms of the "significance audit", which is the second topic searched, we found only 10 results (i.e., articles and proceedings papers) between 1983 and 2020. We found that the research fields that these studies are part of were fewer than those for the "quality audit". Most of them can be found in the Business Finance field, especially in the USA, for most of the published studies (see Figure 2 above). This clearly indicates that the "significance audit" topic should be studied more in the future.

In order to make use of the main research trends in the present topic, we designed a bibliometric analysis of the concepts of "quality audit" and "materiality audit" based on the key words from the titles/abstracts of the articles that were published on Web of Science (WoS), using VOSviewer software.



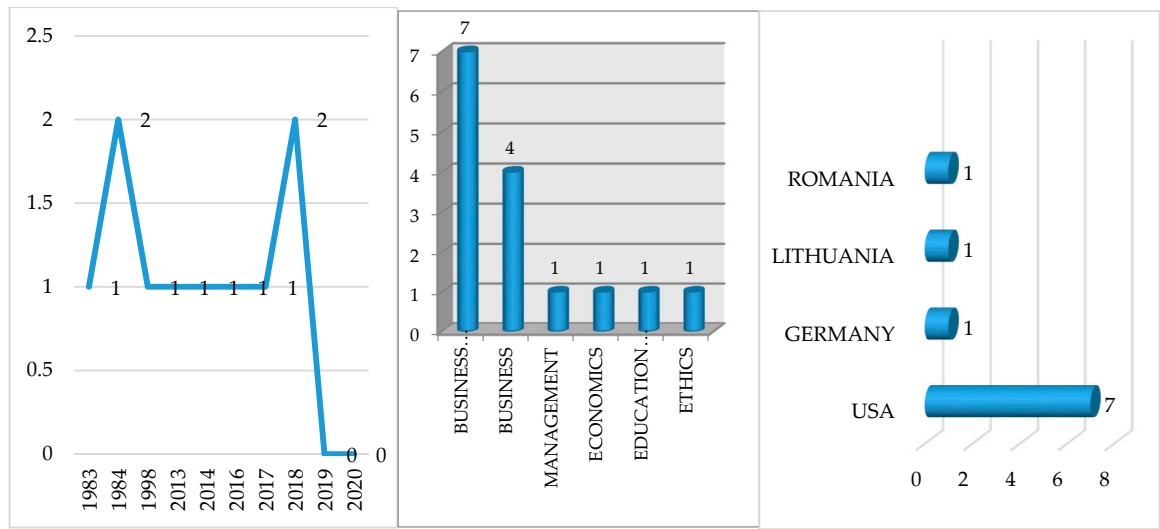

**Figure 2.** The profile of research on materiality audit between 1983 and 2020 in terms of research fields and countries. Source: developed by the authors on the WoS database.

Out of the 2560 terms, the software VOSviewer retrieved 40 terms that met the threshold of at least five occurrences, which were grouped into four clusters (Figure 3). The first two clusters are made up of 12 terms each that are related to terms such as audit, quality and assurance. The third cluster comprises terms such as impact, performance, guidelines, management etc., whereas the latter cluster is made up of seven terms that characterize certain research fields such as public environment, occupational health, agriculture, dairy and animal science.

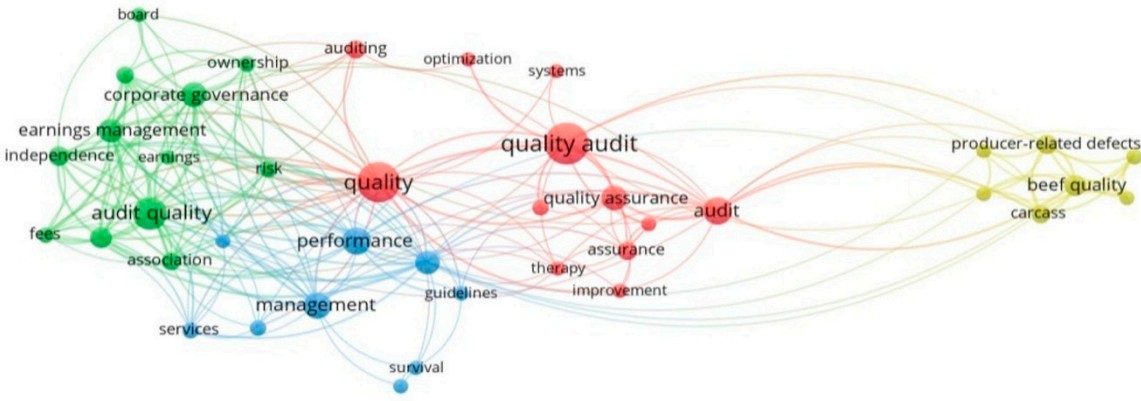

**Figure 3.** Network of keyword co-occurences for quality audit research (2000–2020). Source: developed by the authors using VOSviewer software.

We followed the same procedure for the "materiality audit" (Figure 4). Thus, we obtained three clusters: the first cluster is made of six items, namely, assurance, financial statements, information, judgments, perception and tolerance; the second cluster is geared towards earnings management, tax expense, share prices etc.; the last cluster connects audit materiality with tolerable misstatement and professional judgment.

One can state that the studies belonging to the international literature on "quality audit" and "materiality audit" are to be found in similar contexts, taking into account the fact that both analyses share certain items (e.g., earnings management, assurance). The research on "quality audit" covers more different research fields than that focusing on "materiality audit". Moreover, we noticed that there was a small number of studies that were published on Web of Science that focus on "materiality audit". This is the reason why our research fills a gap that needed to be clarified.

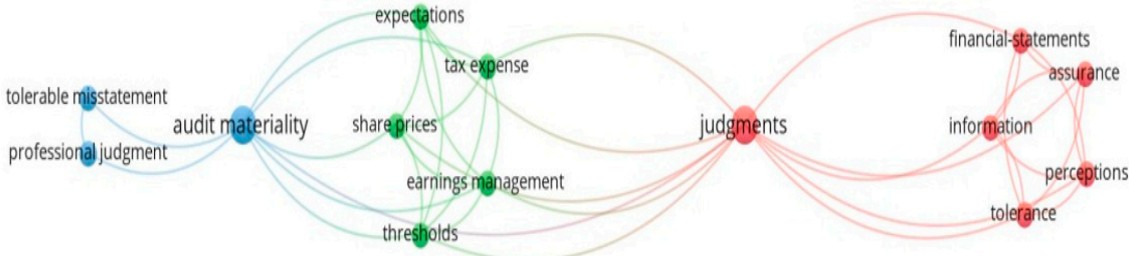

**Figure 4.** Network of keyword co-occurences for materiality audit research (1983–2020). Source: developed by the authors using VOSviewer software.

*2.2. Audit Risk, Materiality and the Professional Judgment of the Auditor*

The concept of materiality, as far as the area of audit is concerned, is associated with the audit decision or judgment decision [11,35–38], audit quality [39,40], earnings management [41–44], audit risk [45–47] and the importance of the client [48,49].

On the other hand, according to Messier et al. [20], most of the studies consider significance from a quantitative point of view where the net profit is the most used variable. However, at present, significance in audit is seen differently, and the focus relies more on the qualitative determining factors—the qualitative audit significance [11,43,50,51]—which, from our perspective, is closely related to a company's sustainability.

For example, Blokdijk et al. [35] state that in practice, unclear supervision as well as the importance of focusing on the materiality while examining the auditor decision may provide useful information for the audit process as such and for its quality.

Furthermore, Popa et al. [52] and Robu et al. [53] highlight auditors' professional judgment, ethical judgment as well as professional ethics as key elements in their decision-making process. Moreover, the authors of [53] showed that an auditor's rotation can significantly contribute to a shift in the relevance degree of the value of the data disclosed by Romanian companies that are quoted on the BVB. Nelson et al. [36] believe that audit regulatory entities need to make sure that auditors use both types of approaches and give up the unimportant adjustments that are necessary for quantitative and qualitative estimations of significance.

Dezoort et al. [37] interviewed 160 auditors and showed that, when facing more responsibility and pressure, they become more conservative in terms of materiality judgments, whereas in an opposite situation they display a more relaxed way of thinking. The findings of the above-mentioned study indicate that the level of responsibility is in a positive relationship with the period of time allotted for fulfilling the tasks that are part of the audit mission, in terms of detailed explanations and taking into account qualitative significance factors. The authors arrived at these data as a result of the fact that the majority of the analyzed studies provided more quantitative than qualitative information, which proved, however, to have the same importance both during the planning process in audit and in the way auditors approach the meaning of significance itself.

Later on, Mittendorf [39] reviewed the audit efficiency and erroneous reporting of data concepts by demonstrating that in time the erroneous establishment of audit threshold may lead to a distortion of the information from financial statements, thus manipulating the correct information and fidelity to the economic reality. Actually, one of Mittendorf's [39] conclusions suggests that fostering some more relaxed audit thresholds can be misinterpreted by the stakeholders as having an opposite effect. Creative accounting techniques can be the key element in the erroneous reporting of accounting information, which fall within the scope of audit effectiveness [54]. Pereş et al. [55] showed that managers are prone to make use of creative accounting techniques in order to bring about certain advantages of a strong brand image for their company. Moreover, the authors emphasized that managers value auditors from two perspectives. The first one is about their image, which is associated

by managers with danger and with individuals that spot fraud, whereas the other perspective is that managers trust the auditors' way of thinking and professionalism.

In a study similar to [37], del Corte et al. [50] demonstrated the importance of qualitative significance factors based on using a questionnaire on 473 subjects (i.e., financial auditors) from Spain. The final results showed that the majority of the auditors who were part of the studied group expressed their agreement in regards to the issue of qualified audit reports when financial statements contain uncorrected misstatements that are below the significance levels but are related to the qualitative significance factors included in ISA 450.

From a different perspective, Acito et al. [41] and Keune and Johnstone [42] analyze the way in which errors regarding financial reporting (i.e., restatement or revision) are rectified in order to determine the significant values of the identified errors. The authors provide evidence on whether the significance assessments are influenced by earnings management incentives. This type of approach offers information on the way the significance is evaluated, yet it fails to provide an analysis or details on the management's actual decisions.

Similarly, Legoria et al. [43] analyzed the significance threshold and earnings management by demonstrating that auditors rely more on the quantitative aspects of the significance threshold than on the qualitative ones.

These conclusions can also be found in the work of Commerford et al. [44], who claim that the tracking of real earnings management affects auditors' adjustment decisions in the same way as the existence of a qualitative significance factor.

On the other hand, Eilifsen and Messier [56] argue that it is important to integrate the means of establishing/determining the significance threshold in the manual of accounting practices for both accounting and audit researchers and practitioners, including regulation entities. For example, the findings of the above-mentioned study indicate that the guides that were implemented by the eight analyzed accounting firms in the USA strictly obey the existing accounting standards and comply with the quantitative reference values (i.e., the profit before taxation, total assets, total revenue or turnover, as well as equities) in order to estimate the significance threshold or to use certain qualitative factors.

Ruhnke et al. [57] studied the relevance of the disclosure of the audit threshold when a credit decision is at stake. Starting from experimental research, where credit decisions were analyzed by board members of German banks, the issue arose of reporting certain different values of the significance threshold (calculated based on certain reference quantitative indicators that are currently used within audit practice) in relationship with the creditors' expectations.

The importance of the significance threshold can also be found as part of an internal audit [58], especially in the configuration of audit opinions that provide identified errors that are evaluated both quantitatively and qualitatively. As far as the quantitative evaluation is concerned, the auditor needs to compare the value of the errors with the significance threshold that was identified during the planning stage. In terms of the qualitative evaluation, the auditor needs to rely especially on his/her experience and working knowledge. In order for an internal audit to be efficient not only in correcting errors but also in preventing accounting fraud from happening [59–61], it is necessary to use adequate internal control mechanisms that are capable of resolving critical situations and protecting the interests of all the stakeholders [62–64]. Thus, the model suggested by Petrov [58] helps in determining the degree of compliance with accounting standards and reporting norms, while improving the planning stage of the internal audit at the same time.

The findings of Čular et al. [65] highlight that external auditors mostly rely on the internal audit function when they offer financial advice to management under the guidance of a strong audit committee. However, they rely less on internal audit when the internal audit function provides only an assurance (under the supervision of either a weak or a strong audit committee).

In the present study, in line with de Rooji [12], we focus on adjusting the significance threshold, which is reflected in a decrease in the gap regarding the significance of the misunderstandings that might appear between the beneficiaries of the financial reports and the auditors.

Gold [66] shows that a persistent expectation gap exists with respect to the auditor's responsibilities. The explanations in the ISA 700 auditors' report do not result in a smaller expectation gap.

Aqel [67] states that as a judgmental concept, however, materiality is "susceptible to subjectivity. Furthermore, the absence of auditing standards on the way materiality is determined has highlighted the significance of this issue and indicated that guidance for materiality professional judgments must come from other non-authoritative sources such as empirical research" (see Figure 5).

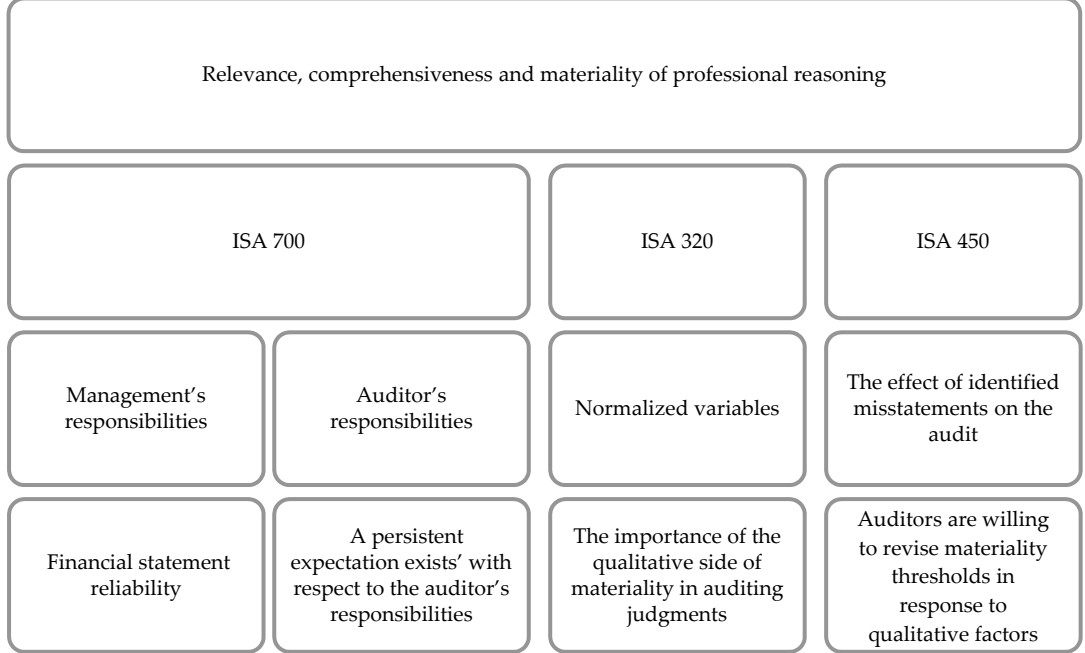

**Figure 5.** Theoretical underpinnings and conceptual framework. Source: authors' creation following [64,65].

Given the situation of our research, we can define the significance threshold in a simpler manner, namely, as the limit zone of errors for financial statements, with one error or more together exceeding the significance threshold while becoming a subject for adjustment.

The concept of significance is a key factor in an audit activity, starting from the very planning stage until the evaluation of the audit tests. Within this context, we can also label the audit expertise in its relationship with the significance threshold as a reliable tool only if the sum of the unadjusted errors in the audit tends towards zero, as there is a direct connection between the expert opinion and the significance threshold.

As a result of examining the specialty literature and in order to argue the reason for our research, we felt it was appropriate to design a brief meta-analysis that will allow us to emphasize the concept of significance and its importance for audit practice (see Table A1).

## 3. Methodology

From a methodological point of view, our study is designed under the form of a series of sequential phases starting from a thorough study of the specialty literature and continuing with the identification of models for sustainable growth of the key indicators that generate optimal economic well-being. Our research endeavor was carried out in parallel with the identification of the current situation and the review models for financial information, which were disclosed by the listed companies on the financial market, viewed in comparison with the volatility of companies' exchange prices.

The study's objectives are as follows:

- O1. To demonstrate the causality relationships between the existence of continuous errors in financial reporting and the inconsistency of audit opinions;

- O2. To demonstrate the economic sustainability of companies that improve their performances based on audit;
- O3. To demonstrate the economic resilience during a period of crisis of companies whose economic sustainability is improved based on audit techniques.

The following working hypotheses were formulated in order to achieve the proposed objectives:

**Hypothesis 1 (H1).** *During an economic crisis, the level of the materiality threshold of audit procedures needs to be at least 30% in order to ensure the validity of the audit solution (see Figure A1).*

**Hypothesis 2 (H2).** *The changes in financial efficiency have a direct impact on the changes in materiality thresholds that are used in audit activity. Professional reasoning of auditors regarding quantitative importance is not only based on the application of general conventional rules, but also on elements related to the size of the assets, the income and the revenues that are reported in a financial statement [38].*

**Hypothesis 3 (H3).** *The improvement of audit techniques contributes to the validation of qualitative accounting information, as demanded by the stakeholders, and generates sustainable development. If the significance threshold is not established correctly then the risk increases, appropriate procedures will not be performed, and financial statements will not meet the requirements of stakeholders [15,68].*

*3.1. Sample*

From a structural point of view, the methodology comprises the conceptualization of a model that is based on the principle of economic analysis regarding series of significant data. Consequently, the entire group of companies listed in the BVB, BET Plus category was considered. Certain inclusion and exclusion criteria were applied for these companies as follows:

The inclusion criteria consisted of (a) those listed companies in the BET Plus category; (b) those companies that are present on the stock market in the last 10 years. The exclusion criteria consisted of (a) the newly listed companies in BET Plus; (b) those companies that have been dormant on the market during the analyzed period.

The analysis of the group was followed by the exclusion of 5 companies out of 39. There were only 34 listed companies that had been reporting regulated financial statements in Romania in the past 10 years, i.e., 2009–2018.

From the point of view of the fields of activity, the listed companies belong to various sectors such as the mining industry, leather industry, pharmaceutical industry, metallurgical industry, aerospace industry, energy sector, finance and banking sector, brokerage sector as well as specialized medical services. The companies' dynamic value of turnover in the past reporting year exceeds 10 million euros, while the number of employees working in the 34 analyzed companies is over 60,000 persons, with an average unit of employees of over 1800 individuals.

*3.2. Measures*

In order to design the QATRMT model (i.e., rapid test for the re-evaluation of the materiality threshold), we will define the following financial parameters:

3.2.1. The Predicted Credit Worthiness Based on Maximizing the Gross Profit Rate in the Altman Model

The calculation of the credit worthiness was done according to the formula given below:

$$CW = \frac{GP}{T} * 100 \tag{1}$$

where:

CW—credit worthiness;
GP—gross profit;
 T—turnover.

The scaling of the credit worthiness indicator in order to validate the sustainable economic growth is established at a minimum 5% annual progress ratio. Whatever falls under this ratio means that the company fails to fulfil all the sustainability criteria, namely, there are vulnerable points of the audit procedures in the past year that are directly connected to these criteria. This situation calls for a rise in the materiality threshold.

In dynamics, the average credit worthiness is expressed according to the formula given below:

$$\overline{CW_i} = \frac{\sum_{t=1}^{n} \frac{GP_{i_t}}{T_{i_t}}}{\sum_{t=1}^{n} t} * 100 \tag{2}$$

where:

$\overline{CW_i}$—the average credit worthiness;
i—the number of analyzed companies, $i \in [1, 34]$;
t—the number of financial periods corresponding to the financial years from 2010 to 2018.

The inner sustainable growth is established in the database for the correct values of the financial credit worthiness on a (1,0) logic variable, while the inner growth of the significance threshold is established in the database in an inverse relationship with the inner economic growth on a (0,1) logic variable.

The matrix of the sustainable test, in a direct relationship with the vulnerability of audit opinions and in an indirect relationship with the necessary adjustment of the significance threshold, is shown below:

$$\begin{pmatrix} \overline{CW}_{ivalid} & 1_s & 0_{ps} \\ \overline{CW}_{iinvalid} & 0_s & 1_{ps} \end{pmatrix} \tag{3}$$

where:

$\overline{CW}_{ivalid}$—the average of the valid credit worthiness ratio;
$\overline{CW}_{iinvalid}$—the average of the invalid credit worthiness ratio.

### 3.2.2. The Representativity of the Operational Income Out of the Total Income

The calculation of the materiality threshold was done based on the formula given below:

$$L = \frac{OI}{TI} * 100 \tag{4}$$

where:

L—the level of representativity of the operational income out of the total income;
OI—operational income;
TI—total income.

The inner efficiency indicator, as a result of the optimization of the representativity level of the operational income as part of the total income for the validation of efficient economic and operational management strategies, was established for a 95% maximum annual progress ratio. Whatever exceeds this ratio means that the company fails to fulfil the efficiency criteria, namely, based on a direct relationship one can identify vulnerabilities in terms of the quality of financial reporting that relate to the validation of the manager's report based on quick audit tests. Consequently, one needs to test both

the internal audit system's reliability and the internal auditor's reports. This situation will indirectly lead to an increase in the level of significance threshold as a result of the economic vulnerability.

In dynamics, the average efficiency ratio is expressed based on the formula given below:

$$\overline{E_i} = \frac{\sum_{t=1}^{n} \frac{OI_{i_t}}{TI_{i_t}}}{\sum_{t=1}^{n} t} * 100 \tag{5}$$

where:

$\overline{E_i}$ —the average of the efficiency ratio;

i—the number of analyzed companies, $i \in [1, 34]$;

t—the number of financial periods corresponding to the financial years from 2010 to 2018.

The inner economic efficiency is established in the database for the correct values of the indicator on (1,0) logic variables, while the inner rise in the significance threshold is established based on the direct relationship with the inner economic growth on a (1,0) logic variable.

The matrix of the sustainable test, in a direct relationship with the vulnerability of audit opinions and in an indirect relationship with the necessary adjustment of the significance threshold, is shown below:

$$\begin{pmatrix} \overline{E}_{ivalid} & 1_e & 1_{ps} \\ \overline{E}_{iinvalid} & 0_e & 0_{ps} \end{pmatrix} \tag{6}$$

where:

$\overline{E}_{ivalid}$ —the average of the valid efficiency ratio;

$\overline{E}_{iinvalid}$—the average of the invalid efficiency ratio.

### 3.2.3. The correlation of the Trend Curves of the Degree of Indebtedness and of the Amount of Total Expenditure in a Calendar Year

The calculation of the degree of indebtedness was done based on the formula given below:

$$Indeb = \frac{TD^*}{TE^*} * 100 = \frac{TD_t - TD_{t-1}}{TE_t - TE_{t-1}} * 100 \tag{7}$$

where:

Indeb—the expression in dynamics of the degree of indebtedness in connection with the sustainable use of resources that were reported based on the total expenditure indicator;

$TD^* = TD_t - TD_{t-1}$ = dynamics of debt accumulation;

$TE^* = TE_t - TE_{t-1}$ = the surplus of the sustainable usage of resources, which is shown by the dynamics of the total expenditure indicator.

The inner efficiency indicator as a result of the optimization of the indebtedness level, which is in a close relationship with the sustainable use of resources for the validation of sustainable economic growth, was established at a maximum 80% annual progress ratio. Whatever exceeds this ratio means that the company fails to fulfil the sustainability criteria. In other words, based on a direct proportionality, one will estimate the vulnerabilities in terms of validating the procedures regarding the precautionary principle. This particular aspect requires testing of the reliability of the accounting methods based on the regulated accounting procedures (which are not specified in the accounting practices manual). This indirectly leads to a rise in the significance threshold as a result of preventing the occurrence of fraud and errors.

In dynamics, the indebtedness ratio is calculated according to the formula given below:

$$\overline{\text{Indeb}_i} = \frac{\sum_{t=1}^{n} \frac{\text{TDT}_{i_t} - \text{TD}_{t-1}}{\text{TE}_{i_t} - \text{TE}_{i_{t-1}}}}{\sum_{t=1}^{n} t} * 100 \tag{8}$$

where:

$\overline{\text{Indeb}_i}$—the average of the indebtedness ratio in relationship with the sustainable use of resources;

i—the number of analyzed companies, $i \in [1, 34]$;

t—the number of financial periods corresponding to the financial years from 2010 to 2018.

The inner economic efficiency corresponds in the database to the correct values of the indicator on a (1,0) logic variable, while the inner significance threshold is in a close relationship with the inner economic efficiency on a (1,0) logic variable.

The matrix of the sustainable test, in a direct relationship with the prevention of fraud and mistakes and an indirect relationship with the necessary adjustment of the significance threshold, is shown below:

$$\begin{pmatrix} \overline{\text{Indeb}_i}_{\text{valid}} & 1_e & 1_{ps} \\ \overline{\text{Indeb}_i}_{\text{invalid}} & 0_e & 0_{ps} \end{pmatrix} \tag{9}$$

where:

$\overline{\text{Indeb}_i}_{\text{valid}}$—the average efficiency valid ratio;

$\overline{\text{Indeb}_i}_{\text{invalid}}$—the average efficiency invalid ratio.

### 3.2.4. The Asset Liquidity Test

The calculation of the asset liquidity test was done based on the formula given below:

$$L = \frac{\text{CA}}{\text{FA}} * 100 \tag{10}$$

where:

L—the assets' level of liquidity;

CA—the value of the current assets;

FA—the value of the fixed assets.

The inner liquidity indicator, as a result of the optimization of the representativity level of the current assets in relationship with the fixed assets for the evaluation of the adjustment to the dynamics of the market, was established at a 120% annual minimum progress ratio. Whatever falls under this ratio means that the company fails to ensure the optimal flexibility and adjustment level for the economic dynamics of the consumer market, namely, in a close inverse proportionality, there are certain vulnerabilities regarding the stakeholders' protection against variations in quotations of stocks. Given this situation, a rise in the level of the significance threshold is recommended in case of a decrease in patrimonial liquidity.

In dynamics, the average liquidity ratio is calculated based on the formula given below:

$$\overline{L_i} = \frac{\sum_{t=1}^{n} \frac{\text{CA}_{i_t}}{\text{FA}_{i_t}}}{\sum_{t=1}^{n} t} * 100 \tag{11}$$

where:

$\overline{L_i}$—the average of the optimal liquidity ratio;

i—the number of analyzed companies, $i \in [1, 34]$;

t—the number of financial periods corresponding to the financial years from 2010 to 2018.

The optimal inner liquidity is assigned to the database for the correct values of the indicator on a (1,0) logic variable, while the inner rise in the materiality threshold is assigned to the database in an inversely proportional relationship with the inner economic growth on a (0,1) logic variable.

The matrix of the sustainable test, in a direct relationship with the audit opinion's vulnerability and an indirect relationship with the necessary adjustment of the significance threshold, is shown below:

$$
\begin{pmatrix}
\overline{L}_{ivalid} & 1_l & 0_{ps} \\
\overline{L}_{iinvalid} & 0_l & 1_{ps}
\end{pmatrix}
\tag{12}
$$

where:

$\overline{L}_{ivalid}$—the average valid liquidity ratio;

$\overline{L}_{iinvalid}$—the average invalid liquidity ratio.

### 3.2.5. Limiting Mistakes and Fraud by Using Judicious Means of Establishing Provisions

The calculation of the vulnerability in terms of fraud and errors was done by using the formula below:

$$
F = \frac{Pv}{EQ} * 100
\tag{13}
$$

where:

F—the limitation of mistakes and fraud based on the use of judicious means of establishing provisions;
Pv—the level of the existing provisions at the end of the financial year;
EQ—the value of the equities at the end of the financial year.

The inner security indicator, based on limiting the error and fraud risks for the validation of the reliability of the internal control and the judicious governance of financial management, is established at a maximum 10% annual progress ratio (as a representation percentage of the provisions that are part of the total equity). Anything above this ratio means that the company fails to fulfil the security criteria in terms of error and fraud risks, namely, in a direct proportionality, there are vulnerable points and a rise in the number of further corrections in the balance sheet of audit mistakes. This aspect calls for an immediate rise in the level of the significance threshold as a result of the economic insecurity.

In dynamics, the average security ratio is expressed by the formula given below:

$$
\overline{F}_i = \frac{\sum_{t=1}^{n} \frac{Pv_{i_t}}{EQ_{i_t}}}{\sum_{t=1}^{n} t} * 100
\tag{14}
$$

where:

$\overline{F}_i$—the average security ratio;
i—the number of analyzed companies, $i \in [1, 34]$;
t—the number of financial periods corresponding to the financial years between 2010 and 2018.

The inner security regarding the fraud and error risks is assigned in the database for the correct values of the indicator on a (1,0) logic variable, while the inner rise in the materiality threshold is assigned in the database according to the inversely proportional relationship with the inner economic growth on a (0,1) logic variable.

The matrix of the security test, in a direct relationship with the efficiency of the internal control and in an indirect relationship with the necessary adjustment of the significance threshold, is given below:

$$
\begin{pmatrix}
\overline{F}_{ivalid} & 1_s & 0_{ps} \\
\overline{F}_{iinvalid} & 0_s & 1_{ps}
\end{pmatrix}
\tag{15}
$$

where:

$\overline{F}_{ivalid}$—the average valid security ratio;
$\overline{F}_{iinvalid}$ —the average invalid security ratio.

### 3.2.6. Human Resources Efficiency

The calculation of the indebtedness degree was done based on the formula given below:

$$HRE = \frac{HRE_t^*}{HRE_{t-1}^*} * 100 = \frac{T_t/Ne_t}{T_{t-1}/Ne_{t-1}} * 100 \tag{16}$$

where:

HRE—the expression in dynamics of the efficiency in using human resources, based on a uniform representation of the turnover in connection with the number of employees;
$HRE_t^* = T_t/Ne_t$ = the efficiency in using human resources at a certain moment in time;
$HRE_{t-1}^* = T_{t-1}/Ne_{t-1}$ = the efficiency in using human resources at a certain moment in time.

The inner efficiency for estimating the labor productivity, in order to validate the sustainable economic growth as well validate the manager's report based on quick audit tests, is done at a minimum 100% annual progress ratio. Whatever is under this ratio means that the company fails to fulfil the sustainability criteria, namely, based on a direct proportionality, one can spot certain vulnerable points in terms of sustainable economics. These aspects call for a rise in the significance threshold when the efficiency in using human resources is reduced.

In dynamics, the average efficiency is calculated based on the formula given below:

$$\overline{HRE_i} = \frac{\sum_{t=1}^n \frac{T_{i_t}/Ne_{i_t}}{T_{i_{t-1}}/Ne_{i_{t-1}}}}{\sum_{t=1}^n t} * 100 \tag{17}$$

where:

$\overline{HRE_i}$ —the average of the efficiency of human resources;
i—the number of analyzed companies, $i \in [1, 34]$;
t—the number of financial periods corresponding to the financial years between 2010 and 2018.

The inner economic performance is assigned in the database for the correct values of the indicator on a (1,0) logic variable, while the inner growth of the materiality threshold is assigned in the database according to the inversely proportional relationship with the inner efficiency on a (0,1) logic variable.

The matrix of the sustainability test, regarding the efficiency of human resources and indirectly regarding the necessary adjustment of the significance threshold, is expressed below:

$$\begin{pmatrix} \overline{HRE}_{ivalid} & 1_r & 0_{ps} \\ \overline{HRE}_{iinvalid} & 0_r & 0_{ps} \end{pmatrix} \tag{18}$$

where:

$\overline{HRE}_{ivalid}$—the valid average of the efficiency in using human resources;
$\overline{HRE}_{iinvalid}$—the invalid average of the efficiency in using human resources.

## 4. Results and Discussion

The materiality allows the auditor to estimate the volume of the audit, to estimate the nature of the accounting errors that are identified by the auditors and, finally, to form an opinion on the reliability and accuracy of the accounting records [69].

As shown in the literature review, materiality has been studied more from the point of view of quantitative factors; however, nowadays qualitative materiality has also become a matter of interest.

Our research focused on a group of 34 companies listed in the BET Plus category on the BVB. Figure 6 shows the structure of the group of companies according to the statistical classification of economic activities in the European Community (NACE).

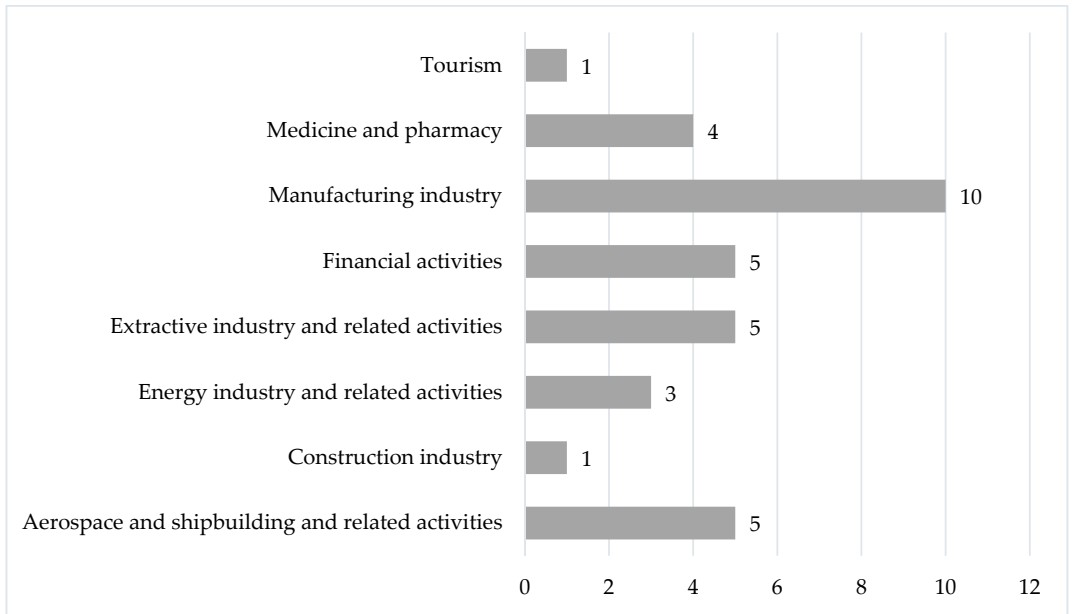

**Figure 6.** The sample structure according to the NACE classification. Source: developed by the authors based on financial data from the Bucharest Stock Exchange (BVB).

There is a homogenous structure among the companies with a more significant weight in the manufacturing industry followed by the energy and extractive industries, financial activities and aerospace and shipbuilding. In our study, we performed a statistical analysis of financial data that were gathered for these 34 companies from 2010 to 2018, and we obtained a database that was later subjected to statistical modeling techniques using SPSS statistical software, version 2019. We also used an Excel model, with which we organized the collected data from financial reports of listed companies. The structure of the database comprises the following fields that were estimated by the authors based on the relevant listings such as: FINSOL (Financial Solvency); FINEFF (Financial Efficiency); INDEF (Indebtedness Effectiveness); FINSEC (Financial Security); ASSLIQUID (Assets Liquidity); SECFE (Security against Fraud and Error); SUSG (Sustainable Economic Growth); INCGST_USUSG (Increasing the Level of Significance Threshold Due to Unsustainable Growth); INCGST_N/AVG (Increasing the Level of Significance Threshold (n/avg); INCGST_AVG (Average Increasing the Level of Significance Threshold), INCGST_2010 (2010 Increasing the Level of Significance Threshold), INCGST_2011 (2011 Increasing the Level of Significance Threshold), INCGST_2012 (2012 Increasing the Level of Significance Threshold), INCGST_2013 (2013 Increasing the Level of Significance Threshold), INCGST_2014 (2014 Increasing the Level of Significance Threshold), INCGST_2015 (2015 Increasing the Level of Significance Threshold), INCGST_2016 (2016 Increasing the Level of Significance Threshold), INCGST_2017 (2017 Increasing the Level of Significance Threshold, INCGST_2018 (2018 Increasing the Level of Significance Threshold); and EQUCAPITALIZ (Equity/Stock market capitalization).

The values of the estimated indicators were modeled two ways. The Auto Regressive Integrated Moving Average (ARIMA) model was used for the analysis in dynamics of the rise in the significance threshold in relationship with the average of the rise in the significance threshold for each and every company from the group. The model had an in-depth statistical significance (i.e., $R^{2 \text{ quotient}} \rightarrow 1$). The statistical tests and the model's configuration are shown below (see Figure 7).

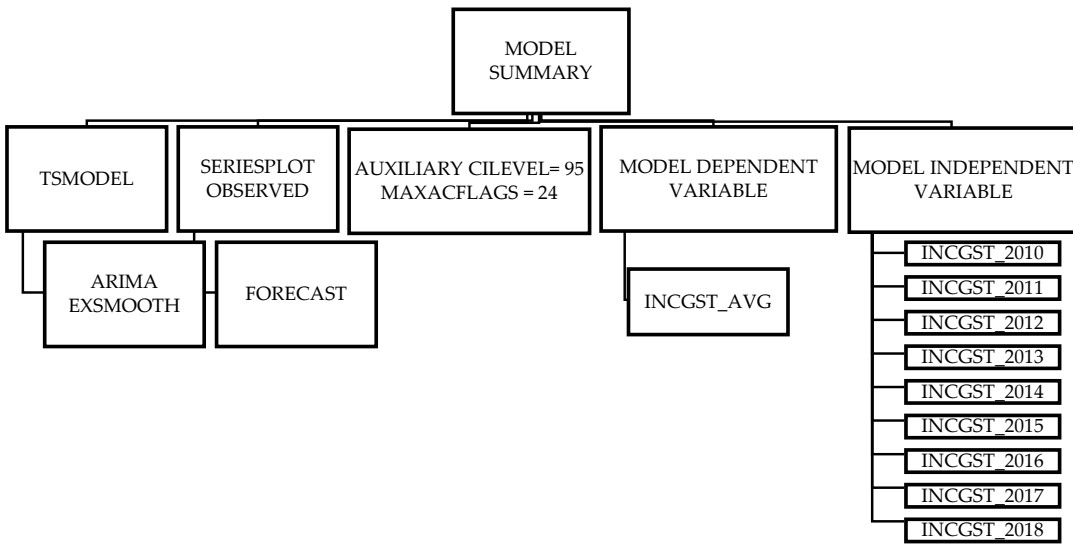

**Figure 7.** Configuration of the Auto Regressive Integrated Moving Average (ARIMA) model.

The statistical test indicates a high level of statistical significance for the dependent variable in relationship with the defined regressors in Table 1 below. There are nine model predictors, and the level of the Sigma coefficient is 0.5. The distribution on the frequency series of the growth of the significance threshold using the ARIMA method fluctuates between a minimum of 14.81% and a maximum of 66.67%. The values of the growing amplitudes of changes made to the significance threshold (+40%) are higher than 50%, while the average value of the changes made to the significance threshold (+30%) is approximate 25%. The model results validate the working hypothesis H1, namely, that during times of economic crisis the level of the significance threshold in audit procedures needs to be at least 30% in order to ensure the validity of the audit solution, and this aspect also validates O1 (see Figure A1).

**Table 1.** Model statistics.

| Fit Statistics | Mean | Minimum | Maximum |
|---|---|---|---|
| Stationary $R^2$ | 1000 | 1000 | 1000 |
| $R^2$ | 1000 | 1000 | 1000 |
| RMSE | $7976 \times 10^{-15}$ | $7976 \times 10^{-15}$ | $7976 \times 10^{-15}$ |
| MAPE | $1146 \times 10^{-14}$ | $1146 \times 10^{-14}$ | $1146 \times 10^{-14}$ |
| MaxAPE | $u3070 \times 10^{-14}$ | $3070 \times 10^{-14}$ | $3070 \times 10^{-14}$ |
| MAE | $4807 \times 10^{-15}$ | $4807 \times 10^{-15}$ | $4807 \times 10^{-15}$ |
| MaxAE | $1421 \times 10^{-14}$ | $1421 \times 10^{-14}$ | $1421 \times 10^{-14}$ |
| Normalized BIC | −63.991 | −63.991 | −63.991 |

| Model | Number of Predictors | Model Fit statistics | Ljung-Box Q(18) | | |
|---|---|---|---|---|---|
| | | Stationary $R^2$ | Statistics | DF | Sig. |
| INCGST_AVG-Model_1 | 9 | 1000 | 28,800 | 18 | 0.051 |

| Model | Number of Outliers |
|---|---|
| INCGST_AVG-Model_1 | 0 |

The diagram of the model is shown in Figure 8.

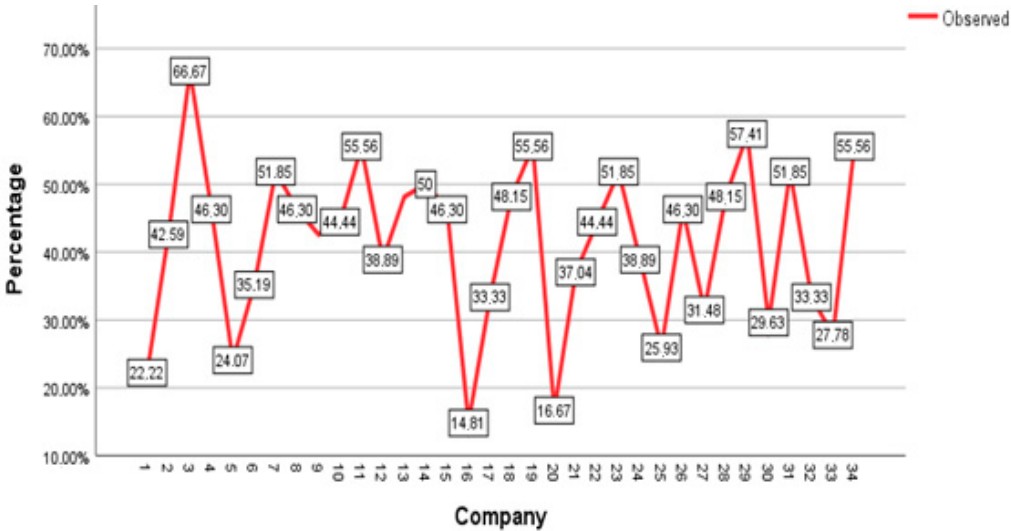

**Figure 8.** The distribution of the dependent variable based on the ARIMA model.

The second trend regarding the modeling of the series of data is the identification of the dependent variable in relation to the regression variables: FINSOL (Financial Solvency); FINEFF (Financial Efficiency); INDEF (Indebtedness Effectiveness); FINSEC (Financial Security); ASSLIQUID (Assets Liquidity); and SECFE (Security against Fraud and Error). We used the ordinary least squares (OLS) method, and the structure of the model is shown in the below figure (Figure 9).

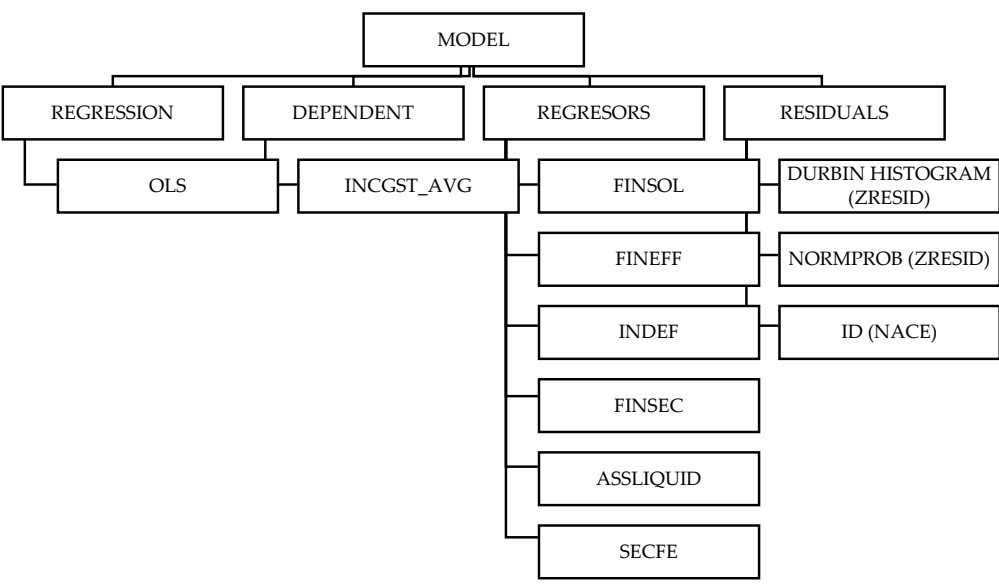

**Figure 9.** The structure of the linear regression model using the ordinary least squares (OLS) method.

Descriptive statistical values of the linear regression model indicate an average of 41% growth of the significance threshold for the dependent variable given the unfavorable dynamics of the financial efficiency indicators, a rise in the degree of indebtedness, and a positive dynamic yet a slow growth of the financial solvency and security (see Table 2). Moreover, the statistical deviations of the frequency series for the dependent variable indicate a degree of homogeneity, which validates the working hypothesis H2, according to which the financial efficiency closely determines the changes in the significance threshold (i.e., O2 is validated).

**Table 2.** Descriptive statistics (N = 34).

| Indicator | Mean | Std. Deviation |
|---|---|---|
| INCGST_AVG | 41.4488% | 12.30927% |
| FINSOL | 7.5644% | 398.50609% |
| FINEFF | −2.3418% | 29.70878% |
| INDEF | −26.9715% | 277.31825% |
| FINSEC | 9.2783% | 129.47452% |
| ASSLIQUID | 10.3665% | 54.79392% |
| SECFE | 17.1440% | 141.01932% |

The table above indicates the dispersion of the results up to a 40% degree of freedom and a degree of alignment of the dispersions of series of regressing data in relationship with the dependent variable that is less than the average of the alignment of the regressing variables. These facts are presented in Table 3.

**Table 3.** Correlations.

| | | INCGST_AVG | FINSOL | FINEFF | INDEF | FINSEC | ASSLIQUID | SECFE |
|---|---|---|---|---|---|---|---|---|
| Pearson Correlation | INCGST_AVG [a] | 1000 | 0.022 | −0.106 | −0.288 | −0.283 | 0.132 | 0.049 |
| | FINSOL | 0.022 | 1000 | −0.274 | 0.102 | 0.144 | −0.012 | 0.002 |
| | FINEFF | −0.106 | −0.274 | 1000 | −0.225 | 0.124 | 0.111 | −0.408 |
| | INDEF | −0.288 | 0.102 | −0.225 | 1000 | 0.293 | 0.118 | 0.338 |
| | FINSEC | −0.283 | 0.144 | 0.124 | 0.293 | 1000 | −0.008 | −0.124 |
| | ASSLIQUID | 0.132 | −0.012 | 0.111 | 0.118 | −0.008 | 1000 | −0.220 |
| | SECFE | 0.049 | 0.002 | −0.408 | 0.338 | −0.124 | −0.220 | 1000 |
| Sig. (1-tailed) | INCGST_AVG | - | 0.452 | 0.275 | 0.050 | 0.052 | 0.228 | 0.392 |
| | FINSOL | 0.452 | - | 0.059 | 0.284 | 0.208 | 0.474 | 0.495 |
| | FINEFF | 0.275 | 0.059 | - | 0.101 | 0.243 | 0.266 | 0.008 |
| | INDEF | 0.050 | 0.284 | 0.101 | - | 0.047 | 0.253 | 0.025 |
| | FINSEC | 0.052 | 0.208 | 0.243 | 0.047 | - | 0.482 | 0.243 |
| | ASSLIQUID | 0.228 | 0.474 | 0.266 | 0.253 | 0.482 | - | 0.106 |
| | SECFE | 0.392 | 0.495 | 0.008 | 0.025 | 0.243 | 0.106 | - |

[a] Dependent variable: INCGST_AVG.

The acquired values from the statistical tests indicate that the analyzed indicators have a high dispersal. Their homogeneity characterizes, at most, 44% of the analyzed sample (see Table 4). However, there are certain valid results of the test for the residual normality, which has a histogramic homogeneity under Gauss' curve.

**Table 4.** Model summary.

| Model | R | $R^2$ | Adjusted $R^2$ | Std. Error of the Estimate | Change Statistics | |
|---|---|---|---|---|---|---|
| | | | | | $R^2$ Change | F Change |
| 1 | 0.440 [a] | 0.194 | 0.014 | 12.22042% | 0.194 | 1080 |

| Model | Change Statistics | | |
|---|---|---|---|
| | df1 | df2 | Sig. F Change |
| 1 | 6 | 27 | 0.399 | 1839 |

| Residuals statistics [b] | | | | | |
|---|---|---|---|---|---|
| Predicted Value | 18.7185% | 52.5181% | 41.4488% | 5.41590% | 34 |
| Residual | −24.30556% | 19.61656% | 0.00000% | 11.05379% | 34 |
| Std. Predicted Value | −4197 | 2044 | 0.000 | 1000 | 34 |
| Std. Residual | −1989 | 1605 | 0.000 | 0.905 | 34 |

[a] Predictors: (constant), SECFE, FINSOL, FINSEC, ASSLIQUID, FINEFF, INDEF; [b] Dependent variable: INCGST_AVG.

The diagram of the residual normality test for the dependent variable is shown in Figure 10.

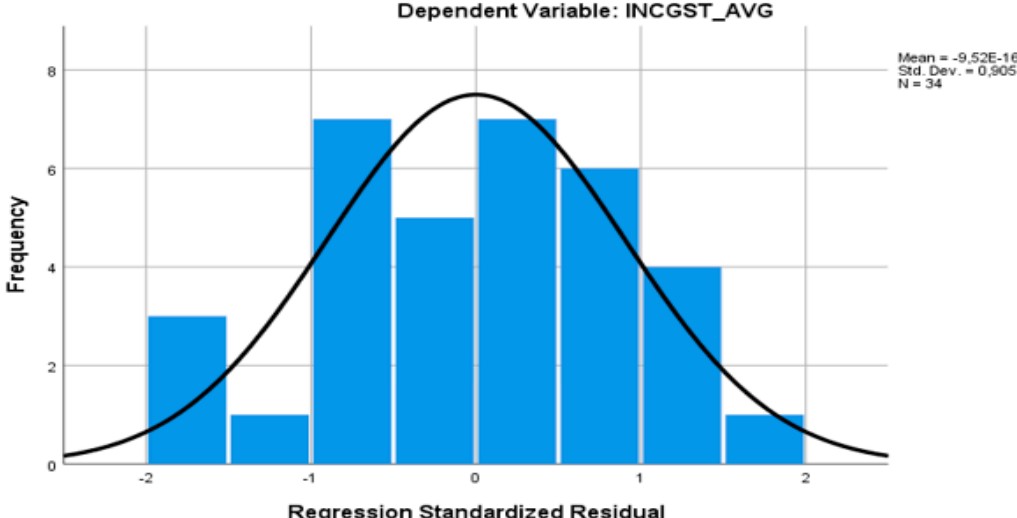

**Figure 10.** Residual normality test: frequency histogram of the dependent variable.

The P-P Plot diagram of the dependent variable displays a representative involution of the frequency series corresponding to the dependent variable on the trend curve (see Figure 11). This demonstrates the model's validity and the homogeneity of the distribution of the dependent variable (O3 validation).

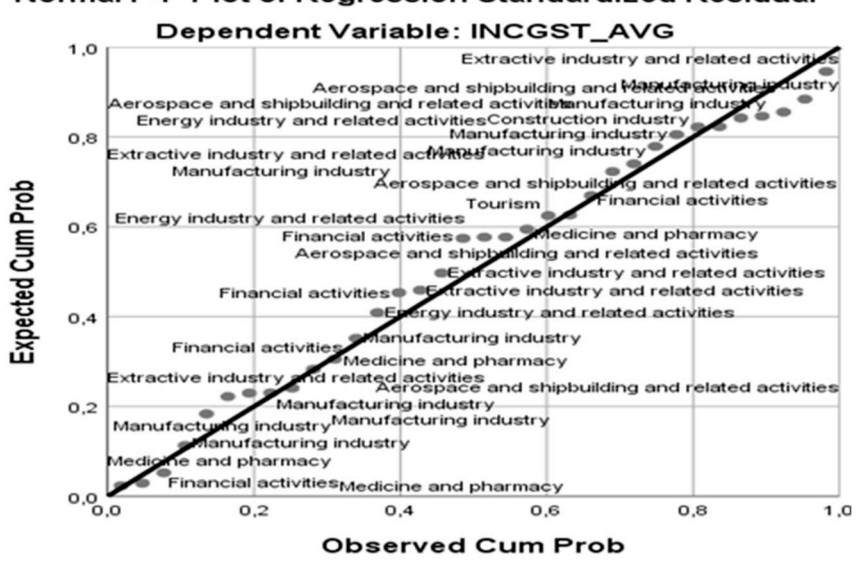

**Figure 11.** P-P Plot diagram of the dependent variable.

Considering the suggested objectives of the present research, we have met these objectives with econometric modeling and have formulated a valid QATRMT model (i.e., quick test for re-evaluating the significance threshold) for the studied phenomenon.

Thus, Figure 12 shows the matrix dispersions of the series of data for SUSG, INCGST and EQUCAPITALIZ, indicators that were created based on the QATRMT model. The dispersions confirm the objectives of our research (O1–O3) and confirm our hypotheses (H1–H3), as defined in the Methodology section, based on the statistical significance level of the data.

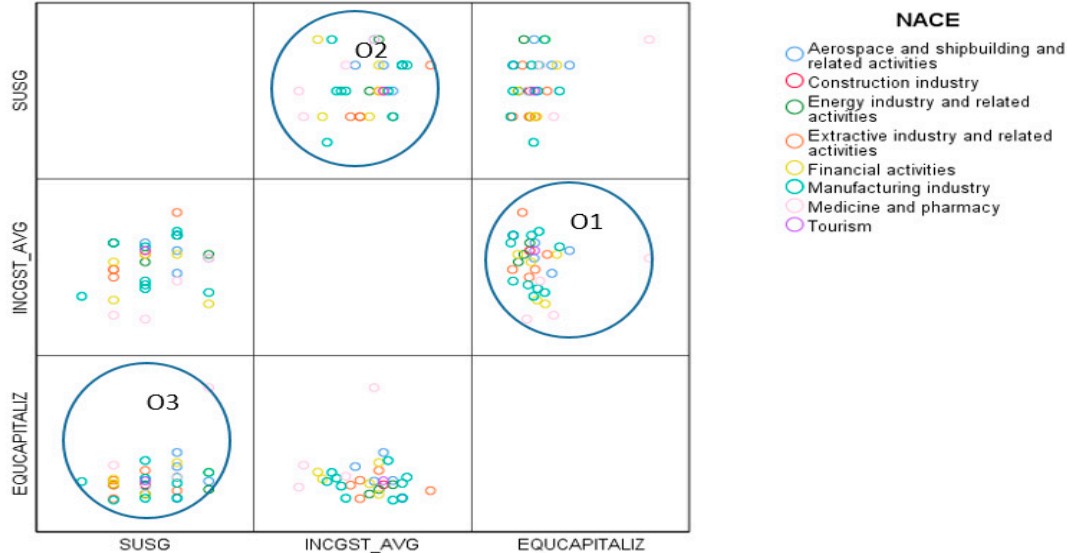

**Figure 12.** The confirmation of our objectives and tested hypotheses.

The present study has developed an econometric model for readjusting significance threshold levels through quick audit tests used on sustainable companies. In order to fulfill the purpose of the present study, we calculated a number of financial indicators that are specific to ten distinct sectors, based on a sample of 34 Romanian companies that are listed on the BVB, in order to identify the nature of the accounting errors that were identified by auditors in order to formulate an opinion on the reliability and accuracy of the financial reporting.

In spite of the fact that the issue of the quality of an audit has been extensively researched and debated both in an academic environment and by practitioners, it never ceases to be a complex and difficult to quantify concept. Consequently, several proxy measurements have been developed. The discretionary methods for measuring the quality of revenues were chosen as a proxy method in our study due to the fact that they capture the slight variations in audit quality that are relevant in terms of taking measures to readjust the significance threshold and in allowing or justifying the sample's fairly limited number. As suggested by the studied specialty literature, the number of clients [70], the size of the audit company [71] and the company's achievements and growth [72] have a significant impact on the audit quality itself. However, so far studies are limited in terms of the evaluation of the impact of financial crises on audit quality [73]. For example, the findings of previous studies have suggested that financial crises might have both a positive and a negative impact on the quality of the revenue and of the audit altogether [74].

Since other specialty studies [75] focus solely on "the issue of magnitude and examine whether financial misstatements that are at or below commonly applied materiality thresholds result in market prices that differ from those resulting from correctly stated information", in our study, based on the analysis of indicators generating financial solvency, we proved that discretionary revenues during a financial crisis may have a severe impact on the quality of the audit itself.

Consequently, our study tested three hypotheses regarding the impact of a financial crisis on the need to adjust the significance threshold (or the quality of the audit) for Romanian companies listed in the BET Plus category on the BVB. The first hypothesis states that during an economic crisis, the significance threshold in terms of audit procedures needs to be at least 30% in order to ensure the validity of the audit solution (as it may lead to an improvement in the quality of audit during the crisis period, which may continue to improve during the post-crisis period). The second hypothesis stipulates that financial efficiency has a direct influence on the changes in the significance threshold, meaning that based on the direct relationship one can estimate the weaknesses in the financial reporting, which leads to the validation of the administrator's report by using the quick audit tests. The third hypothesis

stipulates that the significance of the accounting information is directly linked with satisfying the stakeholders' demands and the sustainable development of the activity of the reporting company, which can be improved by using audit methods. Our suggested model is able to validate the idea that the quality of audit in terms of discretionary revenues has worsened during the financial crisis as a result of the tendency for companies to refrain from disclosing their actual economic situation during times of financial crisis.

Our findings are in line with Lakis & Masiulevičius [15], which state that "materiality level expected by users of financial statements is lower compared to the one, applied by an auditor, therefore, therefore, while calculating materiality the users' opinion has to be considered".

The findings of our study confirm that the quality of audit during periods of financial crisis is questioned if companies disregard the implementation of sustainability criteria. As a result of scaling the financial endpoints that were used in designing the model, we observed that the adjustment of the significance threshold in audit needs to be done during financial crisis periods. Otherwise, the lack of adjustment of the significance threshold will bring about vulnerabilities and risks in terms of audit opinions, such as in the quality of the information in financial reports, the validation of the methods of the prudence principle, protecting investors against volatility variations in listed shares, the level of post-balance sheet corrections for the errors found by the audit, or even vulnerabilities and risks regarding sustainable economics.

## 5. Conclusions

One of the most critical aspects when setting up an audit activity is estimating the risk it poses. This is a frequently underestimated aspect, yet its impact can be a significant one. There are suggestions and tools that can help the auditor to estimate the risk and the significance of the audit activity. However, as far as audit practice is concerned, the evaluation of significance is a subjective estimation in itself due to the fact that in determining and interpreting the significance thresholds, the auditor always needs to make use of his/her professional judgment by taking into account the users' need to be informed of financial situations. Consequently, it is important to know who the users are, what their informational requirements are and, last but not least, how to gather information on the audited company (such as its field of activity, the particular context in which it operates and some information on its lifecycle—expansive, recessive or stable).

The ability of the auditor to identify the significance of errors is a key element of the audit process. An error is viewed as significant when its identification could have brought about change or influenced stakeholders' decisions. The identification of the degree of significance as stipulated by the ISA 320 revision principle allows the auditor, during the revision process, to perform an accurate evaluation of the risks and to react to them based on the audit programs, to logically and rationally identify the nature, timing and length of inspections, as well as to set up a reference point in order to determine the type of opinion he/she has. It must be highlighted that the identification of the significance threshold as stipulated by ISA 320 allows the auditor, given the general audit strategy, to determine both an overall significance threshold (determined for the financial situations) and a specific significance threshold (referring to certain types of transactions). The latter needs to be identified for all the situations when the auditor is convinced that certain economic circumstances may lead to errors that have a smaller value than the general significance but with the same impact on the opinion and the interests of the users of the financial situations.

The present study demonstrates the need to re-evaluate the significance of accounting information both from the point of view of the desynchronization of the global economy during a crisis period and from the point of view of experiencing viable, sustainable growth in the long run. The suggested QATRMT model based on correlated economic indicators, which is used for a series of dynamic data as a result of data gathered from successive financial reports, brings a new perspective on the audit opinion in its relationship with the macro- and microeconomic disruptive factors that indicate the sustainability level both for a particular industry and for the company itself (see Figure A2).

We believe that the findings of our research are useful both for stakeholders and for professionals in the audit field, as we provide an innovative method for quick validation of auditors' opinions, which are used as a reference for stakeholders' decision-making. Moreover, the listed companies are the most important beneficiaries of the results due to the fact that they are given the chance to consolidate their trust in the disclosed financial and non-financial information based on the guarantee that is ensured by the auditor opinion.

From a holistic point of view, we believe that our study will have a significant impact both on practitioners and regulatory entities by shifting the qualitative approaches of the analysis itself towards the key prudential regulations that are stipulated by ISA 320, ISA 450 and ISA 700. The study also highlights the process of refining the information sources that are able to have an impact on the significance, understanding and materiality of business judgments.

Consequently, it must be emphasized that significance does not always have to be understood as a quantitative type of value but needs to be considered from a qualitative point of view too, as there are instances when an error becomes significant due to its effects on the users of the financial situations, irrespective of its absolute value.

One limitation of this study is the small number of variables introduced in the calculation, and thus we propose extending the research through a subsequent scientific paper. Another limitation of our research is in the analyzed time period, which is because in the context of the COVID-19 pandemic it was impossible to collect data for 2019 at the time of the research because the financial reporting in Romania was postponed until the end of July 2020. In addition, the sample size can be considered a limitation of our research. In spite of the fact that the data used in our study covered the full range of economic and financial indicators for entities listed in the BVB BET Plus section, we believe that increasing the sample size through the inclusion of other listed economic entities may contribute to the improvement of the solutions offered by our study. The limitations of our study point us toward future directions of research. The QATRMT can be applied by auditors in readjusting significance threshold levels during critical times, aspect that can contribute to the sustainable development of companies.

**Author Contributions:** Conceptualization, V.G. and M.S.; methodology, V.G. and M.-L.Z.; data curation, M.S. and S.-M.T.; writing—original draft preparation, V.G., M.-L.Z. and M.S.; formal analysis, investigation and visualization, D.M., S.M. and M.-S.C.; supervision and validation, V.G. and M.-L.Z. All authors have read and agreed to the published version of the manuscript.

**Funding:** This research received no external funding.

**Acknowledgments:** This work was supported by project POCU 125040, entitled "Development of tertiary university education to support economic growth—PROGRESSIO", co-financed by the European Social Fund under the Human Capital Operational Program 2014–2020.

**Conflicts of Interest:** The authors declare no conflict of interest.

# Appendix A

**Table A1.** Synthesis of the main studies and their impact on the research field.

| Authors, Year | Main Concepts | Results | Relevance and Impact on the Research Field |
|---|---|---|---|
| Azzali et al., 2018 [40] | Materiality, internal control and quality audit | The study shows that the most important driver of global quality of audit is the capability to detect significant accounts, and the qualitative risk factors are very important for the significance as such. Moreover, the results show that companies use a range of items, such as the quantitative income statement factors and the balance sheet factors, in order to connect the subsidiaries, accounts and processes coherently with the materiality principle. Finally, it is found that three factors (i.e., identifying subsidiaries, identifying significant accounts and associating accounts with processes) often have a direct relation with Scoping, Planning and Risk Assessment Quality. | High impact of useful findings about the relevance of qualitative and quantitative factors in the assessment of significance. In spite of the fact that it has a regional impact (Italy), it can be adapted to other countries. |
| Commerford et al., 2018 [44] | Qualitative materiality, adjustment decisions and real earnings management | Results show that when auditors observe real earnings management, they perceive these operating decisions as aggressive. They make them perceive management as aggressive, as it ultimately leads to larger suggested adjustments on an unrelated audit difference. | Average impact of contributions made to the literature, showing that there is a connection between real earnings management and auditors' response to management. |
| Vieira et al., 2018 [47] | Materiality and audit risk | The finding boils down to the developed Significance Control Index (ISC) that can be used in auditing and easily understood by both auditors and stakeholders. It offers, on the one hand, transparency, security and flexibility in the implementation process and, on the other hand, a greater accuracy in sorting objects, which are evaluated by public or private companies. | High impact given by the methodology used in the development of the ISC. |
| Choudhary et al., 2017 [38] | Auditor quantitative materiality judgments | The results show that looser materiality is associated with fewer audit hours and lower audit fees, thus supporting the construct validity of this measure. At the same time, looser significance judgments are associated with lower amounts of detected errors and a greater incidence of restatements, thus highlighting the importance of these decisions for financial reporting reliability. | High impact given by the results of this study, which links significance judgments with the financial reporting quality. This fact suggests the existence of a significant economic relationship between loose significance thresholds and the incidence of restatements. |
| Ramalho and Pais, 2015 [46] | Materiality and audit risks | As far as quantitative significance is concerned, it was found that there were partial decreases in the level of the significance index in both the planning and the execution, in spite of the fact that the same benchmarks continue to be mostly used with the total assets by auditors. In terms of qualitative significance, there is evidence that there have been changes since the 2008 financial crisis in terms of an increase in the number of factors and the use of qualitative and quantitative factors in determining the significance of the audit. | The high impact of this study lies in the analysis of the effects of the 2008 financial crisis. We are trying to analyze the effects of the current crisis caused by the COVID-19 pandemic. |

**Table A1.** *Cont.*

| Authors, Year | Main Concepts | Results | Relevance and Impact on the Research Field |
|---|---|---|---|
| Popa et al., 2013 [51] | Materiality and qualitative factors | The results indicate that there is a significant correlation between the level of significance and the business sectors that the audited companies are part of, the auditor's experience in the field and the longevity of the relationship with the client. At the same time, the study emphasizes that there is no correlation between the level of significance and the stakeholders' needs in the financial statements nor the management objectives. | High impact due to the fact that the study is based on a sample from Romania, determining the significance level in the audit by taking into consideration qualitative factors. |
| Budescu et al., 2012 [45] | Materiality thresholds and audit risks | The findings show that the reduction of significance may increase or jeopardize the effectiveness of the audit. The auditor's work can be achieved through the quality of internal control and as well as by supplementing more traditional audit tests that produce evidence less likely to be biased toward management. | High impact due to the final results that highlight ways of streamlining the auditor's activity. |
| Chen et al., 2010 [49] | Quality audit and client importance | Firstly, the results suggest that institutional auditors' improvements lead to prioritizing of the costs by compromising quality over the economic benefits from important clients. Secondly, the impact of client importance in terms of audit decisions seems to be different for the individual auditor and for the office level. | Average impact based on the effects of the client's importance on the auditor's opinion. This may be a factor influencing the significance threshold in the audit. |
| Li, 2009 [48] | Auditor's independence and client's importance | Results show no significant statistical association between the audit fees, the non-audit fees, or the total fees and the going-concern opinions in 2001. Instead, in 2003 results show a positive association between the audit (and total) fees and the going-concern opinions. The non-audit fees continue to be separated from the going-concern opinions. | Average impact based on the relationship between the auditor's independence and the client's importance, which can influence significance in audit. |
| Ng and Tan, 2007 [11] | Qualitative materiality thresholds and audit adjustment decisions | The findings indicate that the improvement of the salience of a qualitative significance factor increases the auditors' propensity to book the audit difference, but only for auditors with lower qualitative significance thresholds. At the same time, the findings suggest that the lack of attention given to the qualitative factor may, in part, explain why auditors waive such audit differences. | High impact given by the results of the study in terms of the ambiguities surrounding the materiality thresholds, which refer to the qualitative significance factors. |

Source: authors' compilation.

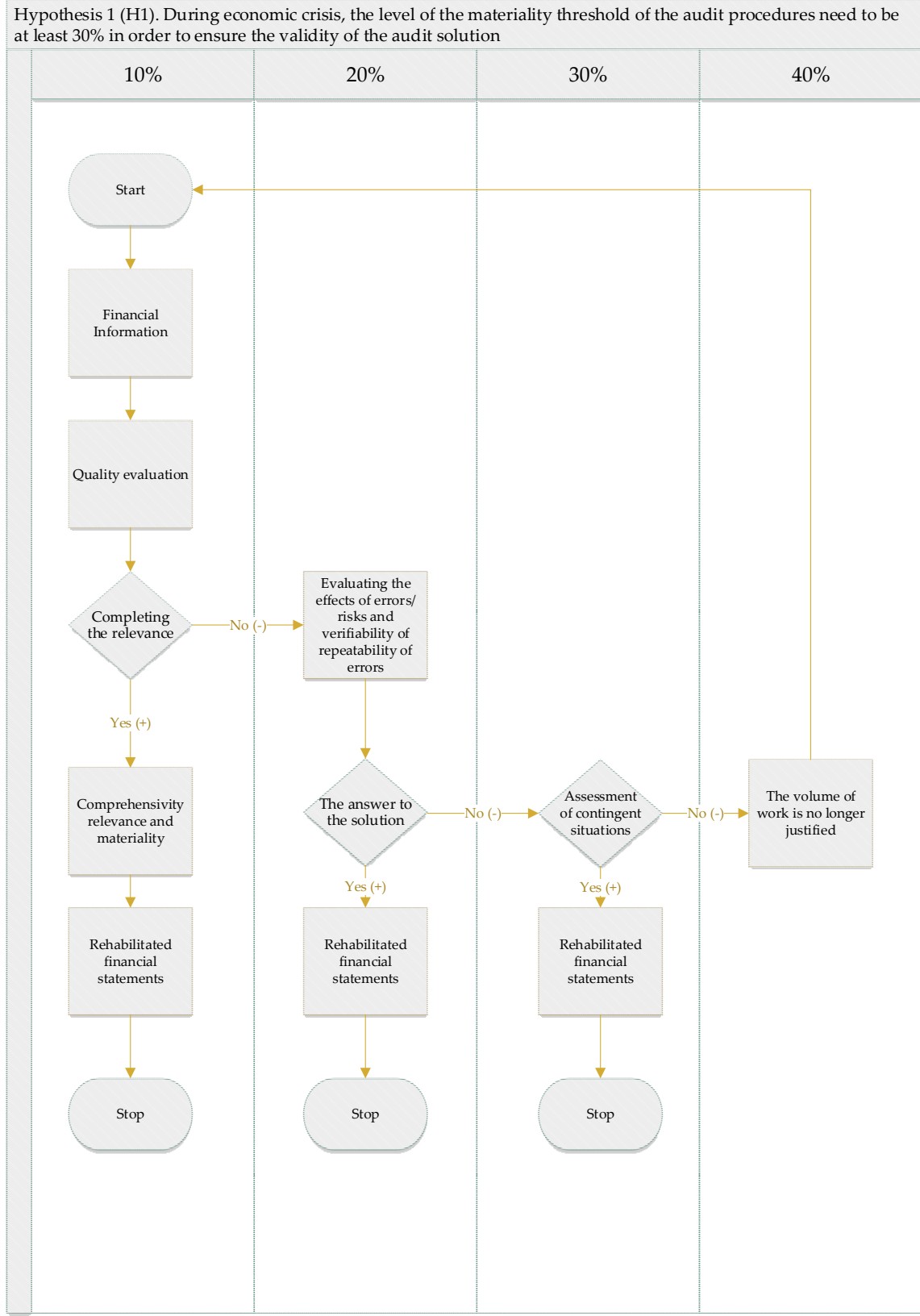

**Figure A1.** Significance thresholds in audit: scenario analysis.

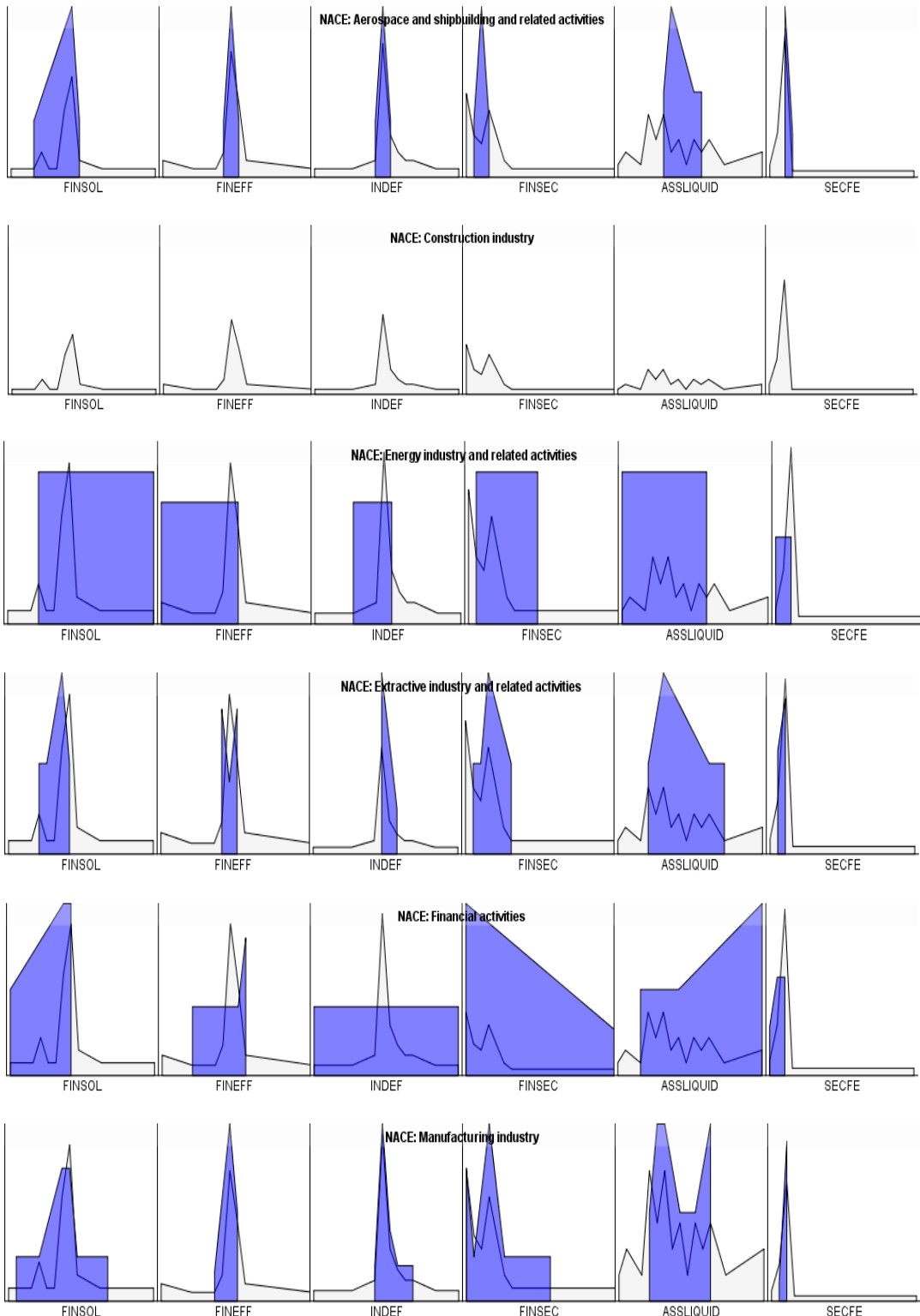

**Figure A2.** *Cont.*

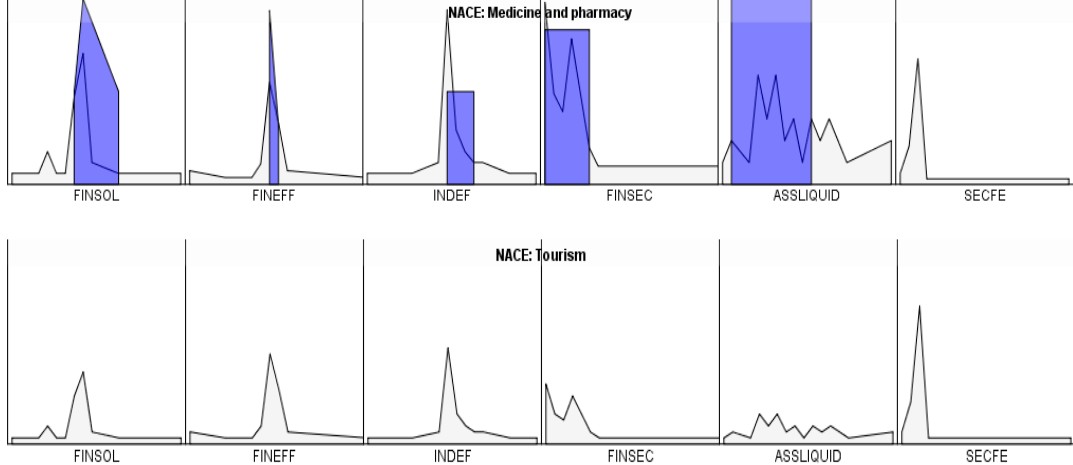

**Figure A2.** The distribution diagram of the sustainability level at company and industry level.

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
