# Peer review of "Econometric Model for Readjusting Significance Threshold Levels through Quick Audit Tests Used on Sustainable Companies"

_sustainability, doi:10.3390/su12198136_

Round 1
Reviewer 1 Report
The comments are attached

Author Response
Dear reviewer,
Please find attached the answers to your requests.

Reviewer 2 Report
- State the key research results in abstract, after the line 30.
- After the line 287, cite this paper-smilar country background. This paper show external auditor reliance on internal audit work in strong audit committee condition. Čular, M., Slapničar, S., & Vuko, T. (2020). The Effect of Internal Auditors’ Engagement in Risk Management Consulting on External Auditors’ Reliance Decision. European Accounting Review, 1-22.
- It is necessary to describe the hypotheses in more detail - why do you expect such directions ....
- H1 is problem with 30% :-) who state 30% - I think that 29% or 31% is the better - :-))) you need explain that
- small sample is big problem !!! n=34 - state that as limitation
Author Response

(The authors gave the same response as above.)
